# A Spontaneously Occurring African Swine Fever Virus with 11 Gene Deletions Partially Protects Pigs Challenged with the Parental Strain

**DOI:** 10.3390/v15020311

**Published:** 2023-01-22

**Authors:** Tomoya Kitamura, Kentaro Masujin, Reiko Yamazoe, Ken-ichiro Kameyama, Mizuki Watanabe, Mitsutaka Ikezawa, Manabu Yamada, Takehiro Kokuho

**Affiliations:** 1Exotic Disease Group, National Institute of Animal Health (NIAH), National Agriculture and Food Research Organization (NARO), Tokyo 187-0022, Japan; 2Nippon Institute for Biological Science, Tokyo 198-0024, Japan; 3Pathology and Production Disease Group, National Institute of Animal Health (NIAH), National Agriculture and Food Research Organization (NARO), Ibaraki 305-0856, Japan

**Keywords:** African swine fever virus (ASFV), immortalized porcine kidney macrophage (IPKM), multigene family (MGF), vaccine development

## Abstract

African swine fever (ASF) is an infectious Suidae disease caused by the ASF virus (ASFV). Adaptation to less susceptible, non-target host cells is one of the most common techniques used to attenuate virulent viruses. However, this may induce many mutations and large-scale rearrangements in the viral genome, resulting in immunostimulatory potential loss of the virus in vivo. This study continuously maintained the virulent ASFV strain, Armenia2007 (Arm07), to establish an attenuated ASFV strain with minimum genetic alteration in a susceptible host cell line, immortalized porcine kidney macrophage (IPKM). A mutant strain was successfully isolated via repeated plaque purification in combination with next-generation sequencing analysis. The isolated strain, Arm07ΔMGF, which was obtained from a viral fluid at a passage level of 20, lacked 11 genes in total in the MGF300 and MGF360 regions and showed marked reduction in virulence against pigs. Moreover, all the pigs survived the challenge with the parental strain when pigs were immunized twice with 10^5^ TCID_50_ of Arm07ΔMGF, although viremia and fever were not completely prevented after the challenge infection. These findings suggest that this naturally attenuated, spontaneously occurring ASFV strain may provide a novel platform for ASF vaccine development.

## 1. Introduction

African swine fever virus (ASFV) is a causative agent for African swine fever (ASF), which is a febrile and lethal infectious disease in pigs and wild boars. ASFV belongs to the nucleocytoplasmic large DNA virus group and is the only known member of the genus Asfivirus of the family Asfarviridae. The virion is 175–215 nm in diameter and consists of a genome structure with five layers: nucleoids, a core shell, an inner envelope, a capsid, and an outer envelope from the inside, in this order [1]. The genome is about 190 kbp of double-stranded DNA which encodes 150–167 genes depending on the isolates [2]. ASFVs are currently classified into at least 24 genotypes based on the B646L gene nucleotide sequence [3].

ASFVs originally existed only on the African continent and circulated between warthogs and Ornithodoros ticks. In 2007, a genotype II strain accidentally invaded Georgia from the African continent and spread to Eastern Europe, Russia, and Asian countries, and it recently spread across Caribbean countries [4]. The ASF outbreak over the last decade has caused significant economic losses to the pig industry in endemic countries.

Although about 100 years have passed since the first ASF case in Africa in 1921 [5], no effective vaccine against ASF has yet been of practical use. Thus far, there are two types of attenuated ASFVs as vaccine candidates: recombinant attenuated viruses generated via genetic modification and naturally attenuated viruses obtained via serial passages in infected animals or in cell cultures. However, most attenuated strains still retain weak virulence and often exhibit persistent viremia and subsequent fever and arthritis in immunized animals [6,7,8,9,10,11].

Naturally attenuated ASFVs emerge spontaneously through serial passages in primary cells or culturable cell lines [12]. Some ASFV strains are known to be capable of infecting non-host animal-derived cells, such as Vero cells, and easily decrease virulence, as well as immunogenicity, through the adaptation process [13,14,15]. Additionally, certain ASFV isolates tend to become less virulent when maintained in cultured cells of host animal origin, such as porcine kidney cells [16,17,18,19].

In this study, in order to establish an ASFV avirulent mutant via non-artificial genetic modification, a highly virulent ASFV strain of genotype II, Armenia07 (Arm07), was continuously maintained in an immortalized porcine kidney macrophage (IPKM), facilitating the efficient propagation of ASFV virulent field isolates [20,21], combined with active emerging virus screening at various passage levels via the plaque purification technique and full genome sequencing analysis. During the process, a novel, spontaneously generated ASFV strain with 11 gene deletions in the genome was successfully isolated (Arm07ΔMGF). By examining the strain’s pathogenicity and vaccine efficacy in pigs, this study clearly demonstrated that this new isolate is capable of inducing partial protective immunity against challenge infection with the parental virulent strain.

## 2. Materials and Methods

### 2.1. Cells

IPKM was established by immortalizing porcine primary culture of kidney macrophages with recombinant lentivirus vectors as previously described [20]. IPKM is highly susceptible to field ASFV isolates and cell-adapted ASFV isolates [21,22]. The cells were routinely maintained in growth medium (Dulbecco’s modified Eagle’s medium (Nakali Tesque, Kyoto, Japan)) supplemented with 10% fetal bovine serum (FBS), 10 μg/mL bovine insulin (Merck, Darmstadt, Germany), 25 μM monothioglycerol (Wako, Osaka, Japan)), and antibiotics in cell culture plates and flasks for suspension culture (Sumitomo Bakelite, Tokyo, Japan). Porcine alveolar macrophage (PAM) cells were prepared from 8-week-old Landrace (L), Yorkshire (W), and Duroc (D) cross breed (LWD) pigs as described previously and stored at −80 °C.

### 2.2. Virus

Dr. Sanchez-Vizcaino (Universidad Complutense de Madrid, Madrid, Spain) kindly gifted the virulent ASFV isolate, Armenia2007 (Arm07, genotype II). Arm07 was propagated in PAM or IPKM cell cultures and used for the experiments. The genome of this isolate differs in only 7 nucleotides from Georgia2007 (GenBank accession no. LR743116). All the experiments with ASFV were performed in the biosafety level 3 (BSL3) facility of the institute, accredited by the national authority of Japan.

### 2.3. Serial Passage of ASFV in IPKM

Arm07 propagated in PAM cell cultures was stored in aliquots and used as an original stock of the virulent virus. An amount of 10^5^ HAD_50_ of the virus was initially inoculated into IPKM cell cultures. The supernatant was collected via centrifugation at 10,000× *g* at 4 °C for 5 min when almost all the cells were detached from the cultureware. This supernatant was defined as the first round of passage virus stock. Then, 10 µL of the stock was inoculated into newly prepared IPKM cell cultures (1 × 10^7^). By repeating the process up to the 20th round of the passage, different passage levels of virus stocks were prepared.

### 2.4. ASFV Genome Next-Generation Sequencing

Genome sequencing was performed as described previously [21]. Briefly, viral supernatant was centrifuged at 180,000× *g* at 4 °C for 3 h. The pellets were then resuspended in 100 µL of PBS and treated with 250 U of benzonase nuclease (Merck) at 37 °C for 1 h. The High Pure Viral Nucleic Acid Kit (Roche, Basel, Switzerland) was used to extract viral DNA and applied to next-generation sequencing analysis using Ion PGM^TM^ (Thermo Fisher Scientific, Waltham, MA, USA) according to the manufacturer’s protocol. The viral reads were trimmed using Trimmomatic v0.36.3 [23] and mapped to the original ASFV Arm07 isolate using Bowtie2 v2.3.0 [24], which were performed using the Galaxy web platform [25].

Genomic analysis of a plaque-purified isolate of Arm07ΔMGF was also performed in the same manner as described above. To determine the exact position of genomic deletion, a set of PCR primers (forward: GACTACTTGGTTAGCAATG, reverse: GTGAGTACACCATACTGAAC) was created at the proximal regions of the deletion predicted by the NGS analysis. Then, conventional PCR and Sanger sequencing were performed using t primers.

### 2.5. Plaque Purification

IPKM (5 × 10^6^ cells/well) was dispersed in 6-well cell culture plates. The cells were inoculated with appropriately diluted virus stock volumes at different passage levels, then, incubated at 37 °C for 1 h. After washing three times with PBS, 3 mL of the growth medium containing 1% SeaPlaque™ GTG™ agarose (Lonza, Basel, Switzerland) was added into each well. A total of 3 mL of the liquid medium was overlaid on each well after solidification and incubated until plaques were visible. The viruses that recovered from the isolated plaques were subjected to two more plaque purification rounds.

### 2.6. Growth Kinetics

IPKM was inoculated with ASFV Arm07 or Arm07ΔMGF at a multiplicity of infection (MOI) of 0.1. The cells were washed three times with PBS and maintained in the growth medium after incubation at 37 °C for 1 h. The supernatants were collected daily up to 5 days post inoculation (dpi). Then, IPKM cell cultures were used to measure the 50% tissue culture infectious dose (TCID_50_) of all supernatants.

### 2.7. Animal Experiments

Animal experiments were conducted in compliance with the Regulations for the Care and Use of Laboratory Animals of the National Institute of Animal Health (NIAH), the National Agriculture and Food Research Organization (NARO), the Guidelines for Proper Conduct of Animal Experiments of the Science Council of Japan [26], and the ARRIVE guidelines [27]. The Institutional Animal Care and Use Committee at the NIAH, NARO (approval number 22-007) reviewed and approved the animal study.

Throughout the study, all pigs received weaner/grower feed and had access to water ad libitum. The animals were observed daily for clinical signs and/or welfare impairment. All efforts were made to minimize animal suffering and to reduce the number of animals used.

#### 2.7.1. Pathogenicity

Arm07ΔMGF virulence was assessed using a total of 13 eight-week-old LWD pigs (both male and female), which were obtained from a commercial health-status herd. All pigs were checked for the absence of ASFV via qPCR [28] and of anti-ASFV antibodies using a commercially available kit (ID Screen^®^ African Swine Fever Indirect kit, IDvet, Grabels, France). The pigs were then randomly divided into three groups (*n* = 3, *n* = 5, and *n* = 5 each). The pig groups were inoculated intramuscularly either with 10^5^ and 10^7^ TCID_50_ of Arm07ΔMGF (*n* = 5 each) or with 10^2^ TCID_50_ of the parental strain (*n* = 3). Clinical signs and body temperature were monitored daily for up to 21 dpi. All the pigs were then necropsied by the end of the experimental period. A humane endpoint was considered when pigs markedly reduced their activity and lay down, justifying euthanasia on welfare grounds. Spleens, kidneys, lungs, and gastrohepatic lymph nodes were harvested from dead or euthanized pigs and homogenized in PBS using a Micro Smash homogenizer (TOMY, Tokyo, Japan).

#### 2.7.2. Protection Efficacy

A total of 12 eight-week-old LWD pigs (both male and female) were obtained from a commercial health-status herd. All pigs were checked for the absence of ASFV and anti-ASFV antibodies using the same test described above. The pigs were then randomly divided into four groups (*n* = 3 each). Single or double doses of 10^3^ or 10^5^ TCID_50_ of Arm07ΔMGF were inoculated intramuscularly into the pigs to assess Arm07ΔMGF’s protective efficacy. In double-dose groups, the second immunization was performed 14 days after the initial immunization. Animals inoculated with Arm07ΔMGF were challenged with 10^2^ TCID_50_ of the virulent parental strain Arm07 at 21 dpi for single-dose groups and 28 dpi for double-dose groups. All pigs’ clinical signs and body temperatures were recorded daily for up to 21 dpc in the single-dose group and 14 dpc in the double-dose group. At the end of the observation period, all pigs were necropsied. The same humane endpoint was applied to this experiment as that described above. Spleens, kidneys, lungs, and gastrohepatic lymph nodes were harvested from dead or euthanized pigs and homogenized in PBS using a Micro Smash homogenizer (TOMY).

### 2.8. Quantitative PCR

The High Pure Viral Nucleic Acid Kit (Roche, Basel, Switzerland) was used to extract the viral DNA from whole blood or tissue homogenate. Then, the PCR was performed according to a previous study [28].

### 2.9. Statistics

The viral growth kinetic data were analyzed via Student’s *t*-test with two-tailed analysis to determine the statistical significance of differences.

### 2.10. ELISA for ASFV-Specific Antibody Detection

Sera in ASFV-inoculated pigs were detected at 0, 7, 14, and 21 dpi and 0, 7, 13 (or 14), and 21 days post-challenge (dpc) using the commercially available ID Screen^®^ African Swine Fever Indirect kit (IDvet) according to the manufacturer’s protocol. The optical density (OD) was read at 450 nm on a microplate reader, Nivo (PerkinElmer, Waltham, MA, USA). The S/P ratio was calculated using the following equation: (sample OD-negative control OD)/(positive control OD-negative control OD).

## 3. Results

### 3.1. ASFV Isolation with Genetic Mutations

The Arm07 strain was maintained by serial passages in IPKM cell cultures up to a passage level of 20. The isolate’s genome sequencing analysis data at different passage levels of 10 (P10) and 20 (P20) were analyzed using a next-generation sequencer as described above. Coverage of the reads was reduced in some parts of the genome of the virus at P20 compared with the original (Figure 1A). A spontaneous mutant lacking a part of the genome was successfully isolated via intensive screening of the virus at P20 after three rounds of plaque purification in agar cultures of IPKM cells (Table 1). Next-generation sequence analysis data of the isolate indicated that it lacked a region corresponding to the region between 21,940 bp and 36,432 bp of the Arm07 strain, resulting in 11 gene deletions (MGF300-4L, MGF360-8L, MGF360-9L, MGF360-10L, MGF360-11L, MGF505-1R, MGF360-12L, MGF360-13L, MGF360-14L, MGF505-2R, and MGF505-3R) in the MGF region of the reference strain (Figure 1B). No mutations were observed, except for a single amino acid substitution in the CP530R gene reported in our previous study [21]. This virus was named Arm07ΔMGF.

### 3.2. Arm07ΔMGF In Vitro Growth

Growth kinetics analysis of Arm07ΔMGF was conducted to examine the 11 gene deletions’ effects on viral proliferation. The parental virus and Arm07ΔMGF were inoculated into IPKM and PAM cells. The supernatants were collected daily until 5 dpi, then, titrated in IPKM cell cultures. In both cells, Arm07ΔMGF growth kinetics were equivalent to those of the parental virus (Figure 2A,B). These data indicate that the 11 gene deletions did not affect the isolate’s replication efficiency in vitro.

### 3.3. Arm07ΔMGF Pathogenicity in Pigs

Pigs were intramuscularly inoculated with 10^5^ or 10^7^ TCID_50_ of Arm07ΔMGF (*n* = 5, each) to determine Arm07ΔMGF virulence. As a control group, 10^2^ TCID_50_ of the parental Arm07 viruses were inoculated via the same route (*n* = 3). All three animals inoculated with the parental viruses showed high fever (>41 °C) and clinical signs, such as anorexia and diarrhea (Figure 3A). By 8 dpi, all animals were dead or euthanized on welfare grounds. In contrast, the experimental pig groups inoculated with 10^5^ or 10^7^ TCID_50_ of Arm07ΔMGF did not present any symptoms or hyperthermia, and survived until 21 dpi (Figure 3A). In the case of parental virus-inoculated pigs, viremia was detected via qPCR from 4 dpi until the end of the experimental period (Figure 3B). The maximum viral copy number in the blood exceeded 10^9^/mL. In contrast, most ArmΔMGF-infected pigs did not show viremia during the observation period, while some showed viremia temporarily at low levels (Figure 3B). At the time of necropsy, viral genes were detected in all of the organs examined in Arm07-inoculated pigs, but no viral genes were detected in the Arm07ΔMGF-inoculated pigs. These data suggested that Arm07ΔMGF possessed a highly attenuated phenotype in vivo.

### 3.4. Arm07ΔMGF Protection Efficacy

The three pigs in each group were intramuscularly immunized once or twice with 10^3^ or 10^5^ TCID_50_ of Arm07ΔMGF to evaluate the protection efficacy of Arm07ΔMGF, and then, challenged with 10^2^ TCID_50_ of parental Arm07 at 21 dpi for the single-dose group or 28 dpi for the double-dose group. Then, the single-dose and double-dose groups were maintained at 21 and 14 dpc, respectively. All immunized pigs had a fever over 40 °C (Figure 4A) after the challenge with the parental strain. The onset of fever after the challenge was delayed in some pigs in the immunized group compared to the non-immunized group. The maximum body temperature was almost the same in the non-immunized group and the single-dose group, but decreased in the double-dose group (Table 2). Notably, fever persisted for a long period, even in surviving pigs. Viremia was found in all pigs except for a pig immunized with a double-dose of 10^5^ TCID_50_ of Arm07ΔMGF (Figure 4B). Although the viremia level tended to be lower in the immunized groups compared to the non-immunized group, it did not depend on the immunized virus amount or the immunization number (Table 2). The viremia was delayed after the challenge up to 1 day compared to the non-immunized group. In the single-dose group, one of the pigs immunized with 10^3^ TCID_50_ of Arm07ΔMGF survived, and two out of the three pigs immunized with 10^5^ TCID_50_ also survived. In the double-dose group, two pigs received low-dose immunization and all the pigs survived. Splenomegaly and extensive hemorrhage in the gastrohepatic lymph node, which are characteristic ASF lesions, were observed in dead and euthanized pigs, but not in the surviving pigs. However, the ASFV gene was detected in spleens, kidneys, lungs, and gastrohepatic lymph nodes via qPCR, even in the surviving pigs.

### 3.5. Humoral Response in Arm07ΔMGF-Immunized/Challenge Pigs

The ID Screen^®^ African Swine Fever Indirect kit (IDvet, France) was used to examine the antibody responses. In pigs immunized with 10^3^ TCID_50_ of Arm07ΔMGF, no pigs became positive before the challenge with the parental virus (Figure 5). In pigs immunized with 10^5^ TCID_50_ of Arm07ΔMGF, one animal in the single-dose group and all pigs in the double-dose group became positive. The remaining two animals also showed a tendency toward increasing antibodies. In all groups, most of the surviving pigs showed a significant antibody level increase one week after the challenge. In contrast, no or only a slight increase was observed in dead or euthanized pigs. These results indicate that humoral immunity was induced in Arm07ΔMGF immunization.

## 4. Discussion

ASF vaccine development is steadily progressing around the world. Naturally attenuated strains have been studied as a possible ASF vaccine strain that incidentally emerges in the process of repeated viral replication in vitro or in vivo. In this study, a plaque assay and genome analysis combination was employed to actively isolate mutant viruses that arose from serial Arm07 strain viral passages in IPKM cell cultures. As a result, a spontaneous mutant from the virus stock at P20, Arm07ΔMGF, was successfully isolated and lacked 11 genes (MGF300-4L, MGF360-8L, MGF360-9L, MGF360-10L, MGF360-11L, MGF505-1R, MGF360-12L, MGF360-13L, MGF360-14L, MGF505-2R, and MGF505-3R) in the genome’s MFG region, which has never been reported before (Table 1, Figure 1A,B).

Cackett et al. reported on the transcription initiation sites of various ASFV genes [29]. The promoter sequences of MGF300-1L and MGF505-4R genes located in the region flanking this deletion were not lost in Arm07ΔMGF, suggesting that the function of both genes was not affected. The possibility of the generation of unexpected ORFs arising from the deletion was considered to be unlikely due to the absence of functional initiation codons or promoters in the proximal region.

ASFV-G-ΔMGF has the most similar genetic deficiencies among the genotype II strains, and lacks 8 out of the 11 genes mentioned above (MGF360-10L, MGF360-11L, MGF505-1R, MGF360-12L, MGF360-13L, MGF360-14L, MGF505-2R, and MGF505-3R) [30]. Of the different genotypes (genotype I), the Pr4Δ35 strain was reported to have genetic defects in 8 out of the 11 genes (MGF360-9L, MGF360-10L, MGF360-11L, MGF505-1R, MGF360-12L, MGF360-13L, MGF360-14L, and MGF505-2R) [31]. Both viruses were created using genetic modification technology and, as far as we know, Arm07ΔMGF is the only isolate that was obtained during the natural attenuation process.

Vero cell-adapted ASFV-G-V110 and 293T cell-adapted ASFV-P121 strains showed a marked decrease in proliferation in porcine macrophage cultures [13,14]. In general, changes in cell tropism may reduce virulence, as well as virus immunogenicity. In contrast, Arm07ΔMGF proliferated in PAM and IPKM cells, which have macrophage-like properties, as well as in the parental strain (Figure 2A,B). Therefore, naturally generated mutations that emerge via serial passages in IPKM cell cultures may not lead to dramatic changes in cell tropism.

In our previous study, we reported that no amino acid substitution in the Arm07 genome was found after 15 rounds of passage in IPKM cultures, except for a single amino acid mutation in the CP530R gene [21]. However, in the present study, a mutant virus with a partial genomic deletion arose after 20 rounds of passage. The rapid disappearance of the mutant virus after a further 10 rounds of passage (in a total of 30 passages) was confirmed via NGS analysis of the virus fluid harvested at a passage level of 30 (data not shown). These findings suggest that Arm07ΔMGF appeared spontaneously and transiently during serial passages in IPKM cell cultures and was not able to be sustained for a long period.

As demonstrated in this study, pigs inoculated with 10^5^ or 10^7^ TCID_50_ of Arm07ΔMGF developed only a few clinical signs (Figure 3A). The viral persistence in the blood ended by 7 dpi and the titers were lower when compared with the parental one (Figure 3B). In addition, no viral gene amplification was detected in the collected organs (spleens, kidneys, lungs, and gastrohepatic lymph nodes) of the pigs dissected at 21 dpi. These results suggest that Arm07ΔMGF was highly attenuated. Interestingly, pigs inoculated with ASFV-G-ΔMGF, which contained similar gene deletion of MGF, MGF505-1R, MGF360-12L, MGF360-13L, MGF360-14L, MGF505-2R, and MGF505-3R [7], exhibited longer viremia than with Arm07ΔMGF. An additional five extra gene deletions in Arm07ΔMGF that are not shared with ASFV-G-ΔMGF may be crucial for further ASFV attenuation. In particular, MGF 360-9L and MGF 360-11L are reported to have type I interferon-inhibitory functions in vitro [32,33]. Investigation of those genes in vivo would provide further insight into the attenuation mechanism.

In the Arm07ΔMGF inoculation case, the dose and the injection number correlated well with the protective efficacy level against the challenge with the parental virus. However, most of the surviving pigs were unable to avoid fever and viremia caused by the parental virus infection (Figure 4A,B). Conversely, all pigs injected with even single dose of ASFV-G-ΔMGF survived throughout the experiment, and only 50% of the pigs developed fever; however, viremia was observed in most surviving pigs, as also observed in [7]. On the other hand, Rathakrishnan et al. generated a mutant with an eight gene deletions (ΔMGF) and examined its vaccine efficacy [30]. However, unexpectedly, the protective efficacy of Arm07ΔMGF was higher than that of ΔMGF, as demonstrated in the previous study [30]. Considering that the ΔMGF mutant possessed three additional MGF genes, MGF300-4L, MGF360-8L, and MGF360-9L, some of these three genes may be important in acquiring immunity against ASFV infection.

Generally, serial passages of virulent viruses in less susceptible, mostly non-host animal-derived cells are performed for spontaneous attenuation purposes. However, some pathogens, especially ASFVs, tend to alter cell tropism to the original target host cells, and dramatically decrease immunogenicity during the process. This study demonstrated that it is possible to obtain mutant viruses without changing cell tropism by combining serial in vitro passages, genome analysis, and repetitive plaque purification in IPKM cell cultures. Since IPKM is suitable for artificial ASFV gene modification (our unpublished data), we aim to use these techniques to improve Arm07ΔMGF’s immunogenicity and develop better vaccine candidate strains.

## Figures and Tables

**Figure 1 viruses-15-00311-f001:**
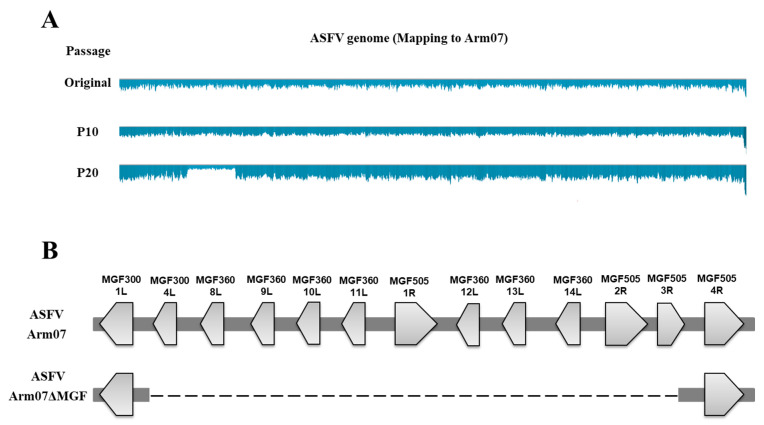
Next-generation sequencing of the passaged viruses and genome structure around the deleted region in Arm07ΔMGF. (**A**) The genomes of the 10th and 20th passed viruses (P10, and P20, respectively) were subjected to next-generation sequencing. Shown in blue is the coverage when mapped to the original Arm07. P20 viruses present coverage reduction in partial genomic regions. (**B**) Schematic gene representation lacking in Arm07ΔMGF. Dotted lines indicate the missing genome region.

**Figure 2 viruses-15-00311-f002:**
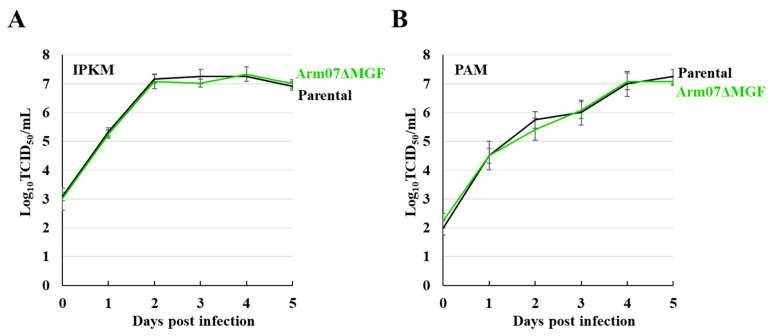
In vitro Arm07ΔMGF and parental Arm07 growth in IPKM and PAM cells. IPKM (**A**) and PAM cells (**B**) were inoculated with Arm07ΔMGF or Arm07 at an MOI of 0.1. The supernatants were collected daily until 5 dpi and subjected to virus titration assays. The data are reported as the mean values with standard deviations for three independent experiments.

**Figure 3 viruses-15-00311-f003:**
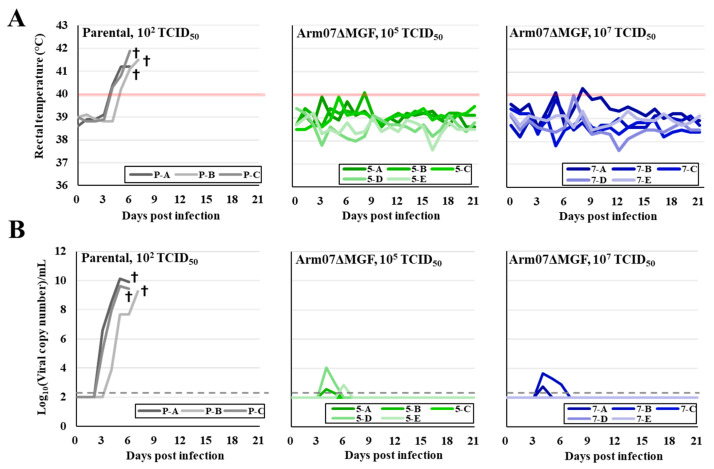
Arm07ΔMGF pathogenicity in pigs. (**A**) Rectal temperature of parental virus- or Arm07ΔMGF-infected pigs. Temperatures above 40 °C (red line) were defined as fever. (**B**) Viral copy numbers in whole blood measured via qPCR. Dotted lines show detection limit of qPCR. Monochrome indicates pigs immunized with 10^2^ TCID_50_ of the parental virus. Green and blue indicate groups immunized with 10^5^ or 10^7^ TCID_50_ of Arm07ΔMGF, respectively. Crosses indicate dead or euthanized pigs.

**Figure 4 viruses-15-00311-f004:**
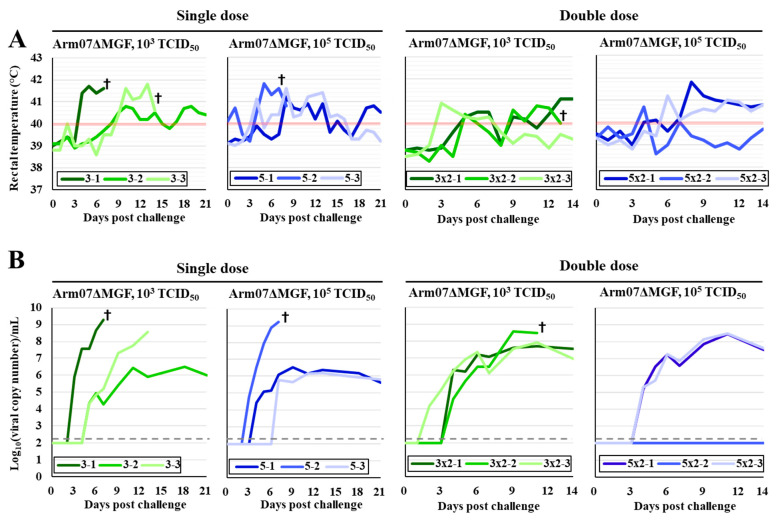
Protective Arm07ΔMGF efficacy in pigs. (**A**) Rectal temperature after challenge with parental virus. Temperatures above 40 °C (red line) were defined as fever. (**B**) Viral copy numbers in whole blood after challenge with the parental virus measured via qPCR. Dotted lines show qPCR detection limit. Green and blue indicate groups immunized with 10^3^ or 10^5^ TCID_50_ of Arm07ΔMGF, respectively. All immunized pigs were challenged with 10^2^ TCID_50_ of parental Arm07 at 21dpi of the single-dose group and 28 dpi of the double-dose group. Crosses indicate dead or euthanized pigs.

**Figure 5 viruses-15-00311-f005:**
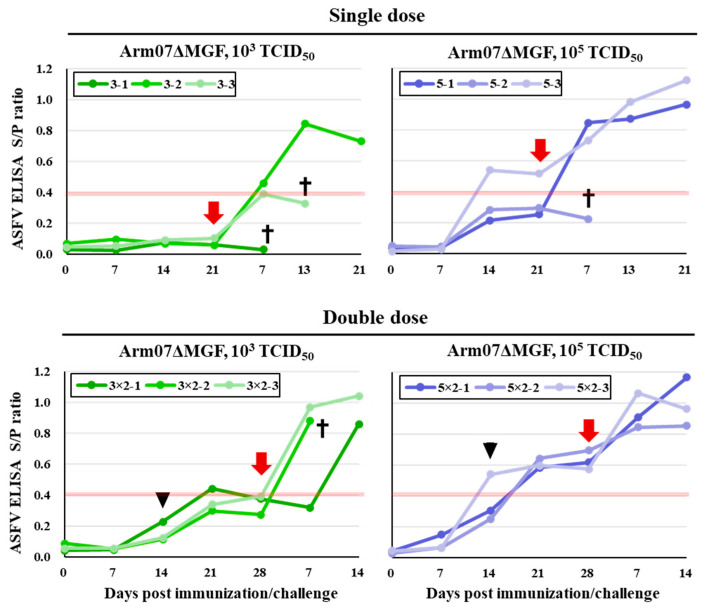
ASF-specific antibody detection. ASF-specific antibodies detected in sera pre-/post-challenge. The red line indicates the lower limit of values that can be determined as antibody-positive. Green and blue indicate groups immunized with 10^3^ or 10^5^ TCID_50_ of Arm07ΔMGF, respectively. The immunization number is displayed above each graph. Crosses indicate dead or euthanized pigs. Black arrowheads indicate the date of the second immunization and red arrows indicate the date of the challenge. The data are indicated as the mean values with standard deviations for three independent experiments.

**Table 1 viruses-15-00311-t001:** Result of the plaque purification.

Round	Number of Plaques	Genetic Identification of the Purified Viruses		
		WT	Deletion	Mix (WT/Deletion)
1	10	1	* 1	8
2	10	0	* 10	0
3	10	0	** 10	0

* One of the purified viruses was used for the next round of purification. ** One of the purified viruses was used for the following experiment.

**Table 2 viruses-15-00311-t002:** Clinical signs and viremia observed in the challenge experiments.

Virus	Dose	Pigs	Surviving or Dead	Maximum of Rectal Temperature (°C)			Maximum Viral Copy Number in Blood (10^n^/mL)		
				Pre	Post	Post Mean	Pre	Post	Post Mean
Parental	single, 10^2^	P-1	D	39.2	41.2	41.5	ND	10.1	9.7
		P-2	D	39.2	41.5			9.3	
		P-3	D	39.6	41.9			9.6	
Arm07ΔMGF	single, 10^3^	3-1	D	39.9	41.7	41.4	ND	9.3	8.1
		3-2	S	40.3	40.8			6.5	
		3-3	D	39.8	41.8			8.6	
	single, 10⁵	5-1	S	41.0	41.2	41.5	ND	6.5	7.3
		5-2	D	40.6	41.8			9.2	
		5-3	S	40.4	41.6			6.2	
	twice, 10^3^	3x2-1	S	40.1	41.1	40.9	ND	7.7	8.1
		3x2-2	D	39.4	40.8			8.6	
		3x2-3	S	40.9	40.9			7.9	
	twice, 10⁵	5x2-1	S	40.0	41.8	41.2	ND	8.4	8.5
		5x2-2	S	40.7	40.7			ND	
		5x2-3	S	40.1	41.2			8.5	

## Data Availability

The data that support the findings of this study are available on request from the corresponding author, Kokuho T.

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
