# Peer review of "A Spontaneously Occurring African Swine Fever Virus with 11 Gene Deletions Partially Protects Pigs Challenged with the Parental Strain"

_viruses, 2023, doi:10.3390/v15020311_

Round 1

Reviewer 1 Report

In this study, Kitamura et al. passaged a highly virulent African swine fever virus strain Armenia07 (genotype II) in an immortalized porcine kidney macrophage (IPKM) and a avirulent mutant with 11 MGF gene deletions was obtained (Arm07ΔMGF). They further evaluated the pathogenicity and efficacy of the mutant as vaccine candidate in pigs, they showed that this new isolate is potent to induce protective immunity against the parental virulent virus challenge. This manuscript is generally well written and provides a new platform for ASF vaccine development, however, several concerns should be addressed.

1.    Next-generation sequencing platform was used to obtain ASFV genome Did the authors confirm the deletion of the mutant by conventional PCR assay?

2.    Please confirm the viral titers at day 0 of the Figure 2.

3.    Lines 170-171, the authors described that P20 of Arm07 strain lacked a region corresponding to the region between 20971 bp and 35459 bp of the Georgia 2007 strain, why they did not compare with the parent virus Arm07 strain?

4.    The authors performed the qPCR according to the paper published by King et al in 2003. The UPL real time PCR that is known to have superior sensibility than that developed by King et al (Gallardo et al., 2015), and actually the latter is also less sensitive than many qPCR kits that currently available.

5.    Figure 3B, the authors are suggested to show Ct value at the same time. 

6.    Other factors such as virus shedding, is necessary to be conducted for comprehensive evaluation of the virus.

Reviewer 2 Report

After carefully Reading the manuscript entitled: “A naturally occurred African swine fever virus with 11 gene deletions protect pigs challenged by parental strain (Ref Viruses-2128622), I recommend Major revisions”

African swine fever is without any doubt the number one threat for the global swine industry. Lack of safe and efficacious vaccines commercially available for global use complicates its control. Therefore, any attempt to increase our knowledge and get closer to this purpose deserves attention.

Despite this reality, the manuscript here presented have severe flaws that should be corrected before publishing in a prestigious Journal such as Viruses.

My MAJOR CONCERN is the lack of accuracy of many statements written in the present work, exemplified in the selection of a TITLE that can mislead the readers, overall of those non-familiarized with the ASF field.

Authors should change the tittle and be more attached to reality:
1- Authors should specify that the 11-gene deletion mutant that they describe (
Arm07ΔMGF),  was not isolated from naturally infected pigs but, it was selected after 20 blind passages in IPKM immortalized cells…far from being a natural host for ASFV

2- Authors should be more modest in their claims since this mutant confers very poor protection (and induces very poor immune responses) compared with many prototypes already available

In my understanding, the tone of the whole manuscript should change, since most of the findings here described are not novel. Authors have neglected to include seminal references describing for example, that most of the Live attenuated viruses used experimentally as vaccines, were obtained using the same methodology here described, based on serially passaging different ASFV strains in immortalized cells (other strains of ASFV and different cells but with similar or even better outcome).

In fact, most of the knowledge we have today, about the mechanisms involved in protection against ASFV, are based on experiments performed withls these naturally attenuated viruses as in vivo mode. Of course, the use of targeted gene recombination allowed extending these studies in a more rational way…

Therefore, passages such as those described in lines 59 to 61: “However, it is mostly unsuccessful to attenuate virulent viruses by repetitive passages in cell cultures [13]”, are false statements. ASF history did not start with the last incursion of the virus in Georgia and most of the viruses circulating before 2007 out from Africa (mostly belonging to the genotype I), were adapted to grow in non-susceptible cells and were successfully used (low passage viruses) as experimental vaccines. It is true however that continuous passage of ASFV in continuous cell lines, lead to extremely attenuated viruses that were not successful as experimental vaccines (multiple references and examples cited in the literature before Georgia07 adaptation to Vero Cells, the only citation in this manuscript)

In fact, conversely to what the authors claim, the results described shoss that:

I believe that the mutant here described clearly demonstrates the opposite than the authors claim. Thus:

1)    Arm07ΔMGF induces weak immune responses and incomplete protection, compared with similar viruses such as GΔMGF, because it lacks too many members of the MGFs compared with the latter; and

2)    That further work should be performed to claim IPKM cells as alternative, at least for large scale vaccine production, since in only 20 passages, the loose at least 11 genes, something never described when using primary swine macrophages, natural ASFV target cells.

This reviewer suggests changing the narrative of the manuscript based on the last sentence of his manuscript (ll322-323): “Since IPKM is suitable for artificial ASFV gene modification, we aim to use these techniques to modify Arm07ΔMGF and develop better vaccine candidate strains”. In my understanding, this is the only real conclusion of the work here presented. The rest, again in my opinion, are overstatements.

These are just examples, authors neglected to mention seminal works describing the relevance of genes adjacent to the genome fragment deleted in Arm07ΔMGF, affected by deletions, duplications, during ASFV adaptation in pigs and in tissue culture cells.

Authors should revise all sections, including title, abstract, introduction, results; material and methods and discussion, citing the many seminal references that they have neglected in almost mot issues tackled.

Thus, the Mat and meth section should be more carefully rewritten, describing in more detail the susceptibility of ASFV to IPKM cells (not only providing references of their origin) and defining much better the blind passages protocol: The amount of virus used in each passage means nothing if the number of cells used are not specified (MOIs should be ideal).

Figure 7 clearly shows that both the wt and the Arm07ΔMGF mutant grow fantastically well in IPKM cells without needing any adaptation. However, and as above mentioned, it is somehow surprising the celerity by which the ASFV changes in IPKM cells (these large deletions are never observed when using Primary swine macrophages, natural ASFV target cells), so this should be more carefully explained. How stable would be this mutant after continuous passage in IPKM?

AUTHORS SHOULD REVISE THE IN VIVO PROTOCOL USED, SINCE SOME TIMES THEY CLAIM THAT PIGS WERE IMMUNIZED WITH 103 or 105 TCID50 of the Arm07ΔMGF AND IN ANOTHER OCCASIONS THEY DESCRIBE THE DOSES  as 105 or 107 TCID50 of the Arm07ΔMGF  

All these questions, and many others should be better addressed in the new revised manuscript.

Reviewer 3 Report

Kitamura et al. describe a genetic modification of the highly virulent African swine fever virus isolate Armenia07 after repeated passages through IPKM cells. The authors describe a large deletion in the left end of the virus genome and characterize the resultant virus in vitro and in vivo. The manuscript is well written and the authors data supports their conclusions. However, some sections are lacking detail and these should be clarified before the manuscript can be accepted for publication. In addition, a growth curve of their deletion virus in primary macrophages should be included as a virus with a similar deletion had a growth defect relative to wildtype (Rathakrishnan et al., 2022).

Line 80: Please define LWD.

Line 105: Please indicate the accession number of Arm07 isolate that was used for the mapping and the specific software used within the Galaxy framework. The method of genome assembly should be included for the final plaque purified ArmΔMGF virus. The data should be uploaded to a publicly available repository.

Line 128: The description of the animal experiment doesn’t meet the ARRIVE guidelines. Points 4 (randomisation) and 5 (blinding) of the ARRIVE Essential 10 are not met and the sex, breed and health/immune status of Item 8 (Experimental animals) are not described.

Line 131. Please provide a more detailed description of the animal experiments, it isn’t clear if these are two separate experiments or one? If only one how where the three pigs that were boosted/challenged chosen? If there are two experiments please define them differently. Both figures 3 and 5 have data on pigs defined as 5.1, 5.2 and 5.3 implying they are the same animals.

Line 171: Please indicate the accession number and version of the Georgia strain used to define the deletion.

Line 170 to 173: It isn’t clear if the authors generated a full genome sequence of their plaque purified virus or not. This is essential and any other changes in the genome (if any) should also be described. In their previous paper the authors describe a single base-pair change in the genome (CP530R gene). Have any new ORFs been generated by the deletion?

Line 181 to 183: Although there is no change in the growth kinetics in IPKMs these are the cells to which the virus has adapted. Therefore, to conclude that the deletion did not affect growth in vitro a comparable growth curve in primary macrophages should be performed. The initial characterization of these cells (Masujin et al.) did not test the growth of attenuated viruses in IPKMs.

Legends for Figure 3 and 4. It is not possible to understand the figures without some intuition and reference to the text. The figure legends should state the doses given to each group and indicate which group is which. At present it relies on the reader guessing that 5.1, 3.2 etc indicates different pigs immunised with different doses. Note this information is included in the legend for Figure 5, please add it here as well.

The discussion misses two main points.

Firstly, the authors need to compare and contrast with their previous work where 15 passages of Arm07 within the IPKMs did not induce any significant changes, whereas in the present study an additional five passages causes a large deletion. The authors should revisit their discussion in Masujin about the suitability of cell types to support the growth of ASFV strains (i.e. live attenuated vaccines) in light of their new data.

The discussion on MGF function should also include reference to and discussion of Rathakrishnan et al., 2022 (https://doi.org/10.1128/jvi.01899-21) which showed a deletion of MGF360-10L, -11L, -12L, -13L, and -14L and MGF505-1R, -2R and -3R attenuated Georgia 2007/1 and only induced 25% protection after challenge. The authors deletion has lost an additional 3 genes and appears to perform better in immunisation/challenge experiments.

Any potential effects of the deletion on the promoters of MGF300-1L and MGF505-4R should be described. See Cackett 2022 https://doi.org/10.1128/jvi.01939-21 for the transcription start sites for these genes. Note this could be included in the results or the discussion.

Line 336: Data statement isn’t correct due to the use of next generation sequencing data.

Round 2

Reviewer 2 Report

I appreciate the efforts made by the Authors

I really think that this manuscript is now, worthy to be shared with the scientific community

Just review/correct the sentence Ll 69-70):
"By examining the strain’s pathogenicity and vaccine efficacy in pigs, this study clearly demonstrated that this new isolate is potent (CAPABLE?) to induce partial protective im munity against the challenge infection with the parental virulent strain".

Reviewer 3 Report

Thank you for the changes you made. I believe the authors manuscript is acceptable for publication in its present form.
